# The Effect of University Students' Individual Innovation and Lifelong Learning Trends on Entrepreneurship Orientation

**Ebru Aykan [1], Gül Karakuş [2],\* and Hande Karakoç [1]**

[1]   Human Resources Management Department, Kayseri University, Kayseri 38280, Turkey;
     aykane@erciyes.edu.tr (E.A.); handeozoren@erciyes.edu.tr (H.K.)
[2]   Department of Strategy Development, Erciyes University, Kayseri 38030, Turkey
\*   Correspondence: gyigitoglu@erciyes.edu.tr

**Abstract:** The aim of this study is to investigate the effects that personality characteristics (PC) of university students, as potential entrepreneurs, have on their entrepreneurship intentions (which are an important indicator of whether or not they will start new initiatives), and to determine the mediating role of lifelong learning tendencies. The study's population consisted of approximately 4000 students at Erciyes University who had relatively high chances of becoming entrepreneurs in total, 924 students were reached. In the research, it was found that the participants' personality characteristics and sub-dimensions had a significant but positive relationship with their entrepreneurial intentions. In the final findings of this study, it was found that lifelong learning tendencies have a partial mediating role in the relationship between the students' personality characteristics and entrepreneurship tendencies. At this point, it was observed that the personality characteristics of individuals affect their entrepreneurship intentions; with the introduction of lifelong learning tendencies to the model, the effect of personality characteristics on entrepreneurial intentions decreased but remained significant.

**Keywords:** personality characteristics; lifelong learning trends; entrepreneurship orientation; university

## 1. Introduction

Efforts towards sustainability, which plays an important role in the economic and social development of countries in a globalizing world, are important in order to achieve successful levels of sustainability. In recent years, it has been observed that the analysis of sustainability studies occurs at an individual, institutional, and universal level. The globalization process has led companies to seek activities that will create a sustainable competitive advantage and value. Entrepreneurs, who are among the most important actors in national economies, implement their ideas and take their businesses into the future with their different perspectives and innovative structures. Entrepreneurship represents an extremely important resource for employment, economic growth, innovation, product and service quality improvement, sustainability, competition, and economic flexibility for both countries and businesses [1]. Entrepreneurship contributes greatly to a country's economic development by promoting innovation, improving economic structures, advancing technology, and creating new jobs, thus playing a key role in increasing economic prosperity, wealth, and sustainability [2]. The changes that the business world in Turkey has experienced in recent years, along with increased competition, decreased public sector employment, and changes in the nature of the workforce, have increased the importance of entrepreneurship development in parallel with economic growth in the information technology (IT) and services sector. In particular, university education, where people's tendencies

towrd entrepreneurship are formed and where these tendencies can be influenced to increase, offers a very important range of entrepreneurship research in Turkey and globally.

On the other hand, knowledge and technology are indispensable elements of our era. Today, thanks to developing technology, access to information has become quite easy. Besides these developments, the existing information and renewal rate have also increased greatly. Therefore, there have been changes in the characteristics of individuals who make up society. The personality characteristics that determine the relationship between the individual and his/her environment, and that express the totality of the features that distinguish him/herself from others, are important factors that guide entrepreneurs in their daily lives [3]. Determining the different personality skills and abilities of students and guiding them in the appropriate fields contributes to the achievement of both individual and corporate success.

With today's renewed, developing, and constantly changing knowledge, the individual has a constant need to learn. This constant change and renewal of information necessitates a continuous learning process. In this process, the concept of lifelong learning comes to the fore. Unlike normal learning, lifelong learning removes the dependence of the learning process on age, time, and space. In order for individuals to achieve the qualifications that are required in this age, it is necessary to gain lifelong skills such as learning to learn. In particular, entrepreneur candidates who are going to operate in a dynamic environment are also changing their competencies to keep up with this constantly changing and developing era. The developing, renewing, and globalizing world, while affecting individuals in many ways, also encourages them to start learning again at any moment. Determining the lifelong learning and entrepreneurship tendencies of university students with different personality characteristics, who are prepared for the business world in particular, is important at this point. Furthermore, in an information society, it is necessary for these students to have their own awareness of individual learning responsibility, to develop their learning skills, to be open to innovation, and to ensure that these skills are sustainable for life. Similarly, entrepreneurship activities, which are expressed as having an invisible effect on the economy, are nourished by innovative, creative, and continuous learning activities, and include formulas for creating sustainable value that are also critical.

Therefore, the aim of this study is to investigate the effect of personality characteristics in university students who are potential entrepreneurs on their entrepreneurial intentionsand determine whether this effect and lifelong learning tendencies are mediated. In this study, the concepts of personality, entrepreneurship, and lifelong learning tendencies will first be explained briefly and the research findings will be included within the framework of the developed model. It is thought that the information obtained as a result of this research will be beneficial for both the literature and practitioners. This is, in particular, due to the necessity of educating creative and innovative individuals, who constitute an important part of a country's competitiveness, with personality characteristics and lifelong learning skills in line with the research results: individuals who are capable of developing their critical and creative thinking, communication, research-inquiry, problem-solving, and entrepreneurship skills. This also benefits institutions. Increasing students' entrepreneurship tendencies increases individuals' success at expressing their ideas, at managing the process of doing business, at adapting quickly to any changes experienced, at making informed decisions, at maintaining motivation in the face of failure, and at developing competencies to find the best solutions.

## 2. Literature Review

### 2.1. Personality Characteristics

The concept of personality, which distinguishes one individual from another and which is expressed as a combination of mental, emotional, and behavioral characteristics, refers to a whole that includes elements such as socio-cultural, family, social, and geographical factors that appear throughout life, along with some innate genetic factors that are important in sustainability studies.

Each individual is different. The concept of personality, which expresses this difference, is expressed as the way in which an individual reacts to or interacts with other individuals [4], and the mental, emotional, and behavioral characteristics that differentiate one individual from another. When the definitions of personality are examined, it can be seen that a person has features that are distinctive from others, the behaviors exhibited by that person gain continuity and stability, and the characteristics that distinguish one person from another are now stereotyped [5]. In the studies on the structure of personality, the permanent characteristics that define the behavior of an individual are identified and named for example as shyness, laziness, ambition, loyalty, and cowardice. Continuous characteristics that define an individual's behavior are expressed as personality characteristics [4].

There are many theories and approaches in the literature that examine personality and characteristics: Sigmund Freud's personality theory, Eric Berne's personality theory, Jung and Adler's personality theory, Horney's personality theory [6], type A and type B personality characteristics, Myers-Briggs type indicators, and five-factor personality characteristics are included. The five-factor personality model from these approaches emphasizes that personality is composed of five different independent dimensions and that these dimensions can be used in the classification of individual differences [7]. Some of the five-factor personality dimensions and characteristics include [4,8–10] those of extrovert people, who are people who love to be together, who are full of energy and who always think well. Openness to experience refers to individuals with imagination, delicacy, curiosity, knowledge, and different interests, and who are adaptable to change, creative, and open to innovation. Emotional stability is a personality trait that is used to describe calm, confident individuals who are self-confident, open to criticism, are patient, and are able to cope with stress. Agreeableness is defined as outspokenness, reliability, and humbleness. Conscientiousness is a personality trait used to define a responsible, orderly, solid character, and disciplined, success-oriented individuals.

## 2.2. Entrepreneurship Intentions

Intentions in general terms can be defined as the representation of future works in the mind. Intentions refer to a person's mental state that directs attention, experience, and behavior toward a particular object or behavior [3]. In this respect, intentions, which can also be called "psychological intuition", are not a simple expectation of actions planned for the future; they also require serious responsibility for the realization of these plans [11].

In general, an entrepreneur is defined as a person who brings together production elements (labor, capital, raw materials, and entrepreneurs) that will create new value for an enterprise and develop economic activities under certain risky conditions [12]. As a broader definition, an entrepreneur has to continually turn to more rational and productive innovations in order to gain superiority over his competitors while realizing and managing economic activity, providing rationality in his/her production and commercial relations, applying new forms of organization and new technologies, developing new products, and introducing new products onto the market. This term also refers to those who lead the markets [13].

Another concept that is closely related to the factors that affect individuals becoming entrepreneurs is entrepreneurship intentions. Entrepreneurship tendencies can be evaluated as a combination of individual and environmental factors, expressing the level of willingness and determination of individuals to create their own business [14]. Therefore, the idea that social factors are important in entrepreneurship intentions is becoming widespread [15]. In this context, entrepreneurship intentions are very simple; they can be defined as an individual's dedication to actions to make an entrepreneurial effort to start his/her own business [16]. In the model proposed by Bird [11], the factors affecting entrepreneurship tendencies and their classification are emphasized; they are economic, social, political, personality, and family characteristics.

## 2.3. Lifelong Learning Tendencies

Today, human lives are longer compared to the past, technological, scientific, and cultural changes in the field of information are constantly and rapidly changing, and learning during a person's life is not limited and constantly reveals the need to develop. In this process, lifelong learning has been a concept that has been emphasized by many institutions and organizations such as United Nations Educational, Scientific and Cultural Organization (UNESCO), Organisation for Economic Co-operation and Development (OECD) and European Union (EU) since the mid-1980s [17]. Lifelong learning tendencies, which are considered to be a process of eliminating negative and insecure thoughts and belief systems, and the discovery of learning tendencies, is particularly important because of the importance of the need for a continuously developing and technologically competent workforce that can compete in global markets in line with the needs of the 21st century, and this is, in the business world, a very popular concept [18]. For this reason, lifelong learning is seen as an important tool in improving the quality of human resources, in bringing them into the economy and making them employable, and in making them countries' most important asset [19]. In particular, EU education programs and the fast-growing business life in of Turkey also necessitate the development of lifelong learning activities [20].

Although complete consensus has not been reached, lifelong learning, which is generally defined as cradle-to-grave learning, is considered to be deliberate, purposeful learning to ensure people's personal development throughout their lives and to improve their quality of life [17]. Since lifelong learning has the purpose of continuously improving the knowledge, skills, and competencies a person should have in every area of his/her life, this form of learning, including diplomatic learning from elementary school to university, is achieved by various state institutions and the training they offer. In addition, learning based on life and work experience is considered to be within the scope of lifelong learning [21]. Therefore, lifelong learning can be realized in schools, universities, at home, at work or anywhere else in society regardless of age, status, or educational level [19,22].

## 2.4. The Relationship Between Personality, Lifelong Learning and Entrepreneurship Tendencies

In the literature, the relationship between individuals' personal characteristics and their entrepreneurial intentions can be seen in many studies [3,23–36]. While some studies show that personality characteristics affect entrepreneurship intentions, others criticize the fact that personality characteristics are based on entrepreneurship research, although personality characteristics are a widely discussed variable in the entrepreneurship literature. Some studies also suggest that there is no relationship between personality characteristics and entrepreneurial performance, as concluded, for example, by Low and MacMillan [37]. Gartner argued that an entrepreneur's average profile cannot be determined because entrepreneurs comprise a heterogeneous group, so it would be more appropriate to emphasize behaviors rather than personality characteristics [38]. Similarly, Keh et al. argued that explaining only individuals' characteristics and business-building behavior cannot explain why people have a business mentality and are willing to take risks [39]. Becherer and Maurer (1999) also argued that studies explaining the concept of entrepreneurship on the basis of individual characteristics are insufficient for explaining entrepreneurship. However, although there is this criticism in the literature, it can be said that personality characteristics still represent one of the most important factors affecting entrepreneurship [36]. In this study, the following hypothesis is proposed:

**Hypothesis 1 (H1).** *The personality characteristics of university students have a positive effect on their entrepreneurship intentions.*

On the other hand, it can be seen that some personality characteristics and their relationship with lifelong learning tendencies have been examined in the literature. Garipagaoglu [40] emphasized the positive relationship between personality characteristics and lifelong learning tendencies, while Eksioglu et al. [41] found that self-efficacy, balance, and openness to experience were relevant

characteristics, and Karaduman and Tarhan [42] and Mulhim [43] found that a characteristic of self-efficacy was positively correlated with lifelong learning tendencies. Based on these findings, the following hypothesis is proposed:

**Hypothesis 2 (H2).** *The personality characteristics of university students have a positive effect on their lifelong learning tendencies.*

Although there are not many studies about lifelong learning and entrepreneurship tendencies in the literature, Gultekin and Erdogan [44] concluded that there is a positive relationship between social entrepreneurship and lifelong learning tendencies. Sezer [45] stated that individuals in need of continuous learning and development are more likely to have entrepreneurship tendencies. Therefore, the following hypothesis is proposed about lifelong learning and the entrepreneurship tendencies of university students who are entrepreneur candidates and thought to have lifelong learning awareness:

**Hypothesis 3 (H3).** *The lifelong learning tendencies of university students have a positive effect on their entrepreneurial intentions.*

The studies in the literature mentioned above found that the personality characteristics and the lifelong learning and entrepreneurship tendencies of university students may be related. However, the literature does not reveal the role of lifelong learning in the relationship between personality characteristics and entrepreneurial tendencies. However, on considering the existence of studies supporting the relationships between the variables, the following hypothesis is proposed:

**Hypothesis 4 (H4).** *Lifelong learning tendencies play a mediating role in the positive effect of university students' personality characteristics on their entrepreneurial intentions.*

## 3. Methodology

### 3.1. Purpose of the Research and Research Model

In recent years, it has been observed that the levels of analysis of sustainability studies are individual, institutional, and universal. The concept of personality, which distinguishes one individual from another and which is expressed as a combination of mental, emotional, and behavioral characteristics, refers to a whole that includes socio-cultural, family, social, and geographical factors that appear throughout life along with innate genetic factors that are important in sustainability studies. Determining students' different personality skills and abilities and guiding them in the appropriate fields contributes to the achievement of both individual and corporate success. Furthermore, in an information society, it is seen as necessary for these students to have an awareness of their individual learning responsibility, to develop their learning skills, to be open to innovation and to ensure that these skills are sustainable for life. Similarly, entrepreneurship activities, which are expressed as having an invisible effect on the economy, which are nourished by innovation, creativity, and continuous learning activities, and which include formulas for creating sustainable value, are also critical. The aim of this study is to investigate the effects the personality characteristics of university students as potential entrepreneur candidates have on their entrepreneurship intentions, representing an important indicator of whether or not they will start new initiatives, and to determine the mediating role of lifelong learning tendencies. The research model is shown in Figure 1 below.

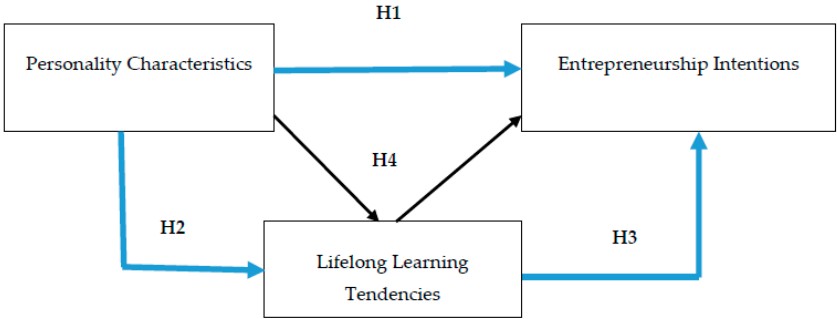

**Figure 1.** Research model.

### 3.2. Data Collection

The scope of this research consisted of students (39,300) continuing their education in technical, health (except the Faculty of Medicine and Dentistry), and social sciences (except the Faculty of Education) fields at Erciyes University. Out of the 39,300 students, 962 questionnaires were returned; however, 923 questionnaires were included in this study.

The study's population consisted of approximately 4000 students at Erciyes University with relatively high chances of becoming entrepreneurs (such as those studying in medical faculties, faculties of education with a high percentage of public works), and 924 students were reached.

The questionnaire, which was created to collect the data, consists of four sections. In the first part, there are 10 questions about the demographic data of the students who completed the questionnaire; in the second part, there are 50 statements to determine the students' personality characteristics; in the third part, there are 17 statements to measure lifelong learning tendencies; and in the fourth part, there are 6 statements to measure entrepreneurial tendencies. In this study, the "5 factors personality characteristics" scale developed by Goldberg [46] the "lifelong learning trends" scale developed by Diker Coskun [47] and the "entrepreneurship tendency" scale developed by Linan and Chen [48] were used. On these scales, 1 represents "strongly disagree" while 5 represents "strongly agree".

### 3.3. Data Analysis

The data obtained from this study were entered into an electronic environment using a statistical package program, and statistical analyses were performed. A Kolmogorov–Smirnov ($n > 50$) test was performed to prove the normality of the distribution, and as $p < 0.05$ for all three scales ($p = 0.000$), the data showed a normal distribution and the kurtosis-skewness values were found to be within the range (+1.5/−1.5). The data were normally distributed and parametric tests were used. The reliability of the scales was calculated using Cronbach's alpha value. In order to test the construct validity of the scales, an exploratory factor analysis was conducted, and whether the previously defined structure was confirmed in the new sample group due to a change in the sample property was examined using confirmatory factor analysis (CFA) and the fit indices were determined. The relationships between the dependent and independent variables were analyzed by calculating the Pearson correlation coefficient, simple regression (H1, H2, and H3) was used to determine the effect of the independent variables on the dependent variables (H4), and hierarchical regression analysis was used to determine the mediation effect (H4).

## 4. Findings

This section presents the findings for the research analysis. First, the demographic data for the managers and their enterprises are included and then the results of the analysis to test the hypotheses are explained.

### 4.1. Demographic Characteristics

Demographic characteristics of the participants are given in Table 1. This study's population consisted of students at Erciyes University (except the Faculty of Medicine, the Faculty of Dentistry and the Faculty of Theology and Education) (*n* = 962). The participants' socio-demographic characteristics are summarized in Table 1. According to the table, the participants were generally male students (51.1%) aged 20 and under (49.5%), studying in the field of social sciences (40.8%), second class (38.4%), and attending a normal education program (69%).

**Table 1.** Demographic characteristics of the participants.

| Variable | Frequency | Percent | Variable | Frequency | Percent |
|---|---|---|---|---|---|
| **Sex** | | | **Age** | | |
| Female | 450 | 48.8 | 20 and under | 457 | 49.5 |
| Male | 472 | 51.1 | Between 21 and 25 | 417 | 45.2 |
| Total | 924 | 100 | 25 and over | 49 | 5.3 |
| | | | Total | 924 | 100 |
| **Area** | | | **Class** | | |
| Technical | 347 | 37.6 | 1. Class | 131 | 14.2 |
| Health | 199 | 21.6 | 2. Class | 354 | 38.4 |
| Social | 377 | 40.8 | 3. Class | 278 | 30.1 |
| Total | 924 | 100 | 4. Class | 132 | 14.3 |
| | | | 5. Class | 28 | 3.0 |
| | | | Total | 924 | 100 |

### 4.2. Validity and Reliability Findings

Cronbach's alpha coefficient was used for the scales' reliability analysis. The reliability of the personality characteristics scale had a Cronbach's alpha value of 0.852, the reliability of the entrepreneurship tendencies scale was 0.782 and the reliability of the lifelong learning trends scale was 0.900.

### 4.3. Factor Analysis for the Scales

The original trait of the personal characteristics scale has five sub-dimensions: extraversion, agreeableness, conscientiousness, emotional stability and openness to experience. An explanatory factor analysis was conducted to determine whether the same dimensions (structure) emerged in terms of this study's data. Factor analysis of the scales is given in Table 2.

It was found that the factor loads of the 26 variables related to the total five dimensions in Table 2 ranged from 0.487 to 0.796, thus satisfying the desired condition. The Kaiser-Meyer-Olkin (KMO), which is the sample adequacy coefficient, was 0.875. Therefore, it can be seen that this research's data structure is suitable for factor analysis. As a result of the factor analysis performed with a Varimax rotation, Table 2 shows that the scale collected under five factors in accordance with the original form. The cumulative total variance explanation rate of the five factors was 52.58%, above the acceptable value of 50%. It was found that for entrepreneurial intentions, factor loadings of one dimension and 6 variables ranged between 0.440 and 0.767, the KMO adequacy coefficient was 0.671, and the cumulative total variance explanation rate was acceptable with 53.56%. Similarly, it was found that for the 17 single-dimension expressions of lifelong learning tendencies collected in one dimension as in the original scale, the total variance explained was 54.78, the KMO adequacy coefficient was 0.934, and the factor loadings varied between 0.451 and 0.760, thus providing the desired conditions. It can be said that the scales were similar to the original scales in terms of structure and had a construct validity.

**Table 2.** Factor analysis for the personality characteristics.

| Variable | Statements | Factor Loading | Factor Validity | Factor Variance |
|---|---|---|---|---|
| Agreeableness | PC10 | 0.793 | 0.813 | 12.19 |
| | PC7 | 0.719 | | |
| | PC11 | 0.681 | | |
| | PC9 | 0.674 | | |
| | PC6 | 0.633 | | |
| | PC8 | 0.611 | | |
| Conscientiousness | PC15 | 0.775 | 0.788 | 11.47 |
| | PC13 | 0.715 | | |
| | PC16 | 0.709 | | |
| | PC17 | 0.676 | | |
| | PC12 | 0.542 | | |
| | PC14 | 0.538 | | |
| Extraversion | PC2 | 0.729 | 0.750 | 10.53 |
| | PC5 | 0.682 | | |
| | PC3 | 0.680 | | |
| | PC4 | 0.646 | | |
| | PC1 | 0.598 | | |
| Openness to experience | PC26 | 0.796 | 0.735 | 9.96 |
| | PC29 | 0.750 | | |
| | PC25 | 0.621 | | |
| | PC27 | 0.511 | | |
| | PC23 | 0.487 | | |
| Emotional Stability | PC18 | 0.763 | 0.704 | 8.41 |
| | PC21 | 0.759 | | |
| | PC20 | 0.685 | | |
| | PC22 | 0.651 | | |
| KMO = 0.877 | *p* = 0.00 | | Total Variance = 52.58 | |

In this study conducted on 924 participating university students, a confirmatory factor analysis was performed to confirm the structures of the scales described above. Whether the previously defined structure was confirmed in the new sample group due to a change in the sample property was examined by confirmatory factor analysis (CFA). The CFA was applied to the five-factor 50-item structure of the personality characteristics scale as a representative of the construct validity. First, there were 24 items with a non-significant *t*-value in the CFA analysis. This item was removed from the scale and the(CFA) model was re-established. When the compliance statistics of the items of the (CFA) model established with 24 items were examined, it was concluded that there were no incompatible items. The compatibility of the items on the scale with the factors formed was confirmed. The fit index values of the personality characteristics scale were $\chi 2/(df)$ 3.50, The Root Mean Square Error Approximation (RMSEA) 0.052, and The Comparative Fit Index (CFI) 0.800. As this is within the range of $0 \leq \chi 2/(df) = 3.50 \leq 5$, it is shown to exhibit acceptable agreement. The RMSEA, CFI, and RMSEA values had an acceptable fit index [49,50].

Similarly, a CFA was conducted on the lifelong learning scale and the CFA was applied to the single-factor 17-item structure of the scale as a representative of construct validity. When the compliance statistics of the items of the CFA model established with 17 items were examined, it was concluded that there were no incompatible items. The adaptation index values of the lifelong learning scale were calculated as $\chi 2/(df)$ 3.35, RMSEA 0.078, CFI 0.948, The Goodness of Fit Index (GFI) 0.986, and the fit index values of the one-factor model of the entrepreneurial intention scale as $\chi 2/(df)$ 4.66, RMSEA 0.085, CFI 0.900, GFI 0.910, and the values were acceptable.

### 4.4. Findings Regarding the Research Hypotheses

The correlation matrix showing the relationships between the variables according to the results of the study is shown Table 3.

**Table 3.** Correlation matrix.

| | | Mean | Standard Deviation | 1 | 2 | 3 | 4 | 5 | 6 | 7 |
|---|---|---|---|---|---|---|---|---|---|---|
| (1) | Personality Characteristics | 3.66 | 0.46 | 1 | | | | | | |
| (2) | Agreeableness | 3.45 | 0.75 | 0.640 * | 1 | | | | | |
| (3) | Conscientiousness | 4.04 | 0.68 | 0.717 * | 0.359 * | 1 | | | | |
| (4) | Extraversion | 3.72 | 0.72 | 0.702 * | 0.267 * | 0.365 * | 1 | | | |
| (5) | Openness to Experience | 3.52 | 0.68 | 0.509 * | 0.081 * | 0.243 * | 0.279 * | 1 | | |
| (6) | Emotional Stability | 4.93 | 0.80 | 0.759 * | 0.454 * | 0.407 * | 0.377 * | 0.304 * | 1 | |
| (7) | Lifelong Learning Tendencies | 4.01 | 0.59 | 0.695 * | 0.435 * | 0.546 * | 0.537 * | 0.265 * | 0.527 * | 1 |
| (8) | Entrepreneurship Intentions | 3.31 | 0.59 | 0.402 * | 0.269 * | 0.203 * | 0.193 * | 0.298 * | 0.403 * | 0.363 * |

\* $p > 0.01$.

In this study, it was seen that personality characteristics and sub-dimensions had above average scores, especially the emotional stability dimension, which had the highest score of 4.93. While the participants' lifelong learning tendencies were determined to be over 4.01, their entrepreneurship tendencies were found to be 3.31. As shown in Table 3, it was determined that the participants' personality characteristics have a meaningful and positive but weak relationship with their entrepreneurship intentions, and a significant positive and moderate relationship with their lifelong learning tendencies. A significant positive but weak relationship was also found between lifelong learning and entrepreneurial intentions.

In this study, a simple linear regression analysis was performed to test the above hypotheses. The analysis and its findings are shown in the Table 4.

**Table 4.** Coefficient table of the regression analysis to determine the effect of personality characteristics on entrepreneurial intentions.

| | Unstandardized Coefficients | | Standardized | *t* | Sigma |
|---|---|---|---|---|---|
| | Beta | Standard Error | Beta | | |
| Constant | 1.436 | 0.143 | | 10.00 | 0.00 |
| Personality Characteristics | 0.515 | 0.039 | 0.399 | 13.23 | 0.00 |

Table 5 shows results of regression analysis to determine the effect of personality characteristics on entrepreneurial intentions.

**Table 5.** Results of regression analysis to determine the effect of personality characteristics on entrepreneurial intentions.

| | R | $R^2$ | Adjusted $R^2$ | Standard Error of the Estimate | F | Sig. |
|---|---|---|---|---|---|---|
| Personality Characteristics | 0.399 | 0.159 | 0.158 | 0.546 | 175.12 | 0.00 |

$p < 0.05$, dependent variable: entrepreneurship intentions.

The descriptive coefficient ($R^2$) in Table 5 is the most common way of measuring the goodness of fit of the linear model. This coefficient shows how much of the change in the dependent variable can be explained by the independent variable(s). This is a good expression of the explanatory power of the regression model. Starting from this, it can be said that part of the change in entrepreneurship intentions such as 0.159 can be explained by the personality characteristics argument in the research model. When the relationship between the variables is examined, the beta value of 0.399 indicates that there is a positive and medium-level relationship between personality characteristics and entrepreneurial intentions. Accordingly, the **H1** hypothesis that "*the personality characteristics of university students have a positive effect on their entrepreneurial intentions*" is accepted.

Table 6 shows coefficient table of the regression analysis to determine the effect of personality characteristics on lifelong learning trends.

**Table 6.** Coefficient table of the regression analysis to determine the effect of personality characteristics on lifelong learning trends.

| | Unstandardized Coefficients | | Standardized | *t* | Sig. |
|---|---|---|---|---|---|
| | Beta | Standard Error | Beta | | |
| Constant | 0.774 | 0.114 | | | 0.00 |
| Personality Characteristics | 0.885 | 0.031 | 0.685 | 6.78 | 0.00 |

Table 7 shows results of the regression analysis to determine the effect of personality characteristics on lifelong learning trends.

**Table 7.** Results of the regression analysis to determine the effect of personality characteristics on lifelong learning trends.

| | R | $R^2$ | Adj. $R^2$ | Std. Error of The Est. | F | Sig. |
|---|---|---|---|---|---|---|
| Personality Characteristics | 0.685 | 0.470 | 0.469 | 0.434 | 819.56 | 0.00 |

$p < 0.05$ dependent variable: lifelong learning tendencies.

In terms of the regression analysis results on lifelong learning tendencies, which are the independent variables of the personality characteristics of the university students participating in the study, personality characteristics are statistically significant ($p = 0.05$) and (β value 0.685) positive. The relationship and $R^2$ value were calculated as 0.470, thus indicating a moderate level of explanatory power. In the light of these findings, the **H2** hypothesis is accepted.

Table 8 shows coefficient table of the regression analysis to determine the effect of lifelong learning tendencies on entrepreneurial intentions.

**Table 8.** Coefficient table of the regression analysis to determine the effect of lifelong learning tendencies on entrepreneurial intentions.

| | Unstandardized Coefficients | | Standardized | *t* | Sig. |
|---|---|---|---|---|---|
| | Beta | Standard Error | Beta | | |
| Constant | 1.940 | 0.125 | | 15.51 | 0.00 |
| Lifelong Learning Tendencies | 0.344 | 0.031 | 0.344 | 11.15 | 0.00 |

Table 9 shows results of the regression analysis to determine the effect of lifelong learning tendencies on entrepreneurial intentions.

**Table 9.** Results of the regression analysis to determine the effect of lifelong learning tendencies on entrepreneurial intentions.

| | R | $R^2$ | Adj. $R^2$ | Std. Error of The Est. | F | Sig. |
|---|---|---|---|---|---|---|
| Lifelong Learning Tendencies | 0.344 | 0.118 | 0.117 | 0.559 | 124.30 | 0.00 |

$p < 0.05$, dependent variable: entrepreneurship intentions.

Similarly, 0.181 of the change in entrepreneurship intentions can be said to be explained by the lifelong learning disposition argument in the research model. When the relationship between the variables is examined, the beta value of 0.344 indicates that there is a positive and medium-level relationship between lifelong learning tendencies and entrepreneurial intentions. According to this study, the **H3** hypothesis that "*the lifelong learning tendencies of university students have a positive effect on their entrepreneurial intentions*" is accepted.

While the regression analysis with the mediator variable can be used for different approaches, the most common one is the causal step approach, also known as the Baron and Kenny (1986) method. According to this approach, some assumptions must be made in order to perform the mediation test. These assumptions are stated below [51]:

- There should be a statistically significant relationship between the dependent variable and the independent variable.
- There should be a statistically significant relationship between the independent variable and the mediating variable.
- There should be a statistically significant relationship between the mediator variable and the dependent variable.
- When the independent variable and the mediator variable are included in the analysis, full mediation can be mentioned if there is a non-significant relationship between the independent variable and dependent variable, and a partial mediation effect can be mentioned if there is a decrease in the relationship between the independent variable and dependent variable.

Table 10 shows results of the regression analysis of the mediator role of lifelong learning trends in the relationship between university students' personality characteristics and their entrepreneurship intentions.

**Table 10.** Results of the regression analysis of the mediator role of lifelong learning trends in the relationship between university students' personality characteristics and their entrepreneurship intentions.

| | Dependent Variable | Independent Variable(s) | | | | | |
|---|---|---|---|---|---|---|---|
| | | B | Std. Error | Beta | *t* | *p* | F |
| **Test 1** Personality Characteristics | Entrepreneurship Intentions | 0.515 | 0.039 | 0.399 | 13.23 | 0.00* | 175.12 |
| **Test 2** Personality Characteristics | Lifelong Learning Tendencies | 0.885 | 0.031 | 0.685 | 28.62 | 0.00* | 819.56 |
| **Test 3** Constant | Entrepreneurship Intentions | 1.333 | 0.146 | | 9.11 | 0.00* | 93.710 |
| Personality Characteristics | Intentions | 0.397 | 0.053 | 0.307 | 7.46 | 0.00* | |
| Lifelong Learning Tendencies | | 0.133 | 0.041 | 0.133 | 3.24 | 0.01** | |

$\Delta R^2 = 0.168$; $p^* < 0.00$, $p^{**} < 0.05$.

In this context, first the relationship between the independent variable (personality characteristics) and the dependent variable (entrepreneurial intention) was investigated and a positive significant relationship was observed. Second, the effect of the independent variable (personality characteristics) and the mediator variable (lifelong learning tendencies) was examined and a positive relationship was

determined. Finally, in order to determine the mediator role (lifelong learning tendencies) mediation variable between the independent variable (personality characteristics) and the dependent variable (entrepreneurial intentions), a regression analysis was added to the analysis as an independent second variable and the mediation effect was evaluated. After the employees' lifelong learning tendencies were added to the model, the effect of personality characteristics on their entrepreneurial intentions decreased from $\beta = 0.399$ to $\beta = 0.307$. It is a fact that the effect of personality characteristics diminishes after the lifelong learning disposition is added to the model, but that this effect is significant shows that a lifelong learning disposition is a partial mediator variable in this relationship. This requires the **H4** hypothesis to be accepted.

## 5. Results, Conclusions, and Recommendations

Sustainability studies are based on the individual's awareness that resources are limited in his/her physical, emotional, social, philosophical, and intellectual life. At the same time, it is emphasized that awareness should be raised with regard to transferring resources to future generations. An individual who is conscious about sustainability can analyze problems systematically in order to understand the environmental, social, and economic aspects of society. At this point, the concept of personality, which is the main player in individuals' behavioral differences, comes to the fore. Many psychologists, instructors, philosophers, and engineers have addressed this complex structure, which includes what constitutes personality, how it manifests, and the personality characteristics that are necessary to train a multi-functional individual. A common theme in all of this research is that personality is related to various factors that form a complex system where changes may cause other unforeseen changes in other factors [52].

Similarly, financial and environmental scandals and problems in recent years have caused social unrest and environmental decay. Individuals should be held responsible for the emergence of these problems. On the other hand, it can be seen that these individuals are fundamentally capable of innovative, creative, and solution-oriented entrepreneurial efforts. Individuals with sustainable entrepreneurial characteristics adopt a holistic business approach that combines economic, social, and environmental values, and implement these practices in this direction. These entrepreneurs provide services using their ability to choose sustainable and innovative practices, to create social networks and to generate financial income. At this point, it is expected that there will be certain personality characteristics that do not define sustainable entrepreneurial behaviors and actions. Personality characteristics are partly developed by innate competences, socialization, and education. These characteristics are also composed of values and beliefs, and play a guiding role in the decision-making processes of entrepreneurs. If these personality characteristics can be identified, it is thought that business, management, and sustainability training will increase entrepreneurial drive.

The biggest problem for university students is finding a job in their specialized areas. Entrepreneurship at this point offers a way out for these young people. It is an alternative for them to start their own business and contribute to the economy with products and ideas that create added value within the scope of their competencies.

In recent years, entrepreneurship has been expressed as an invisible hand in the economy by businesses and countries. It can be seen that both the importance of entrepreneurship and the number of studies on this issue have increased along with a recognition of the need for entrepreneurs who think differently and create sustainable value with their innovative ideas. Entrepreneurship intentions are one of the dimensions of entrepreneurship. Intentions refer to a person's mental state that directs attention, experience, and behavior toward a particular object or behavior. In short, intentions predict an individual's mental state before their behavior. Therefore, it is important to investigate the entrepreneurship intentions of university students, who are potential entrepreneurial candidates.

The aim of this study is to investigate the effect of university students' personality characteristics on their entrepreneurial intentions and to determine whether there is a mediator role in lifelong learning tendencies.

In the research, it was found that the participants' personality characteristics and sub-dimensions had a significant and positive relationship with their entrepreneurial intentions. Past studies on the subject support the findings of the present study [3,23–36]. Although there is a positive relationship between the dimensions of personality and entrepreneurship tendencies [53] openness to emotional balance and experience [54], and extroversion, Zhao et al. [30] found that there is no relationship between the dimensions of emotional balance, while Low and MacMillan [37] concluded that there is no relationship between personality and entrepreneurship tendencies.

A significant, positive and moderate relationship was determined between personality characteristics and lifelong learning tendencies. This result was reported by Garipagaoglu [40] and Eksioglu et al. [41]. Similarly, a significant positive but weak relationship was found between the participants' lifelong learning and entrepreneurship intentions. Gultekin and Erdogan [44] found a positive relationship between social entrepreneurship and lifelong learning tendencies. In the last finding of this study, it was found that lifelong learning tendencies have a partial mediating role in the relationship between the students' personality characteristics and entrepreneurship tendencies. At this point, it was observed that the personality characteristics of individuals affect their entrepreneurship intentions; and with the introduction of lifelong learning tendencies to the model, the effect of personality characteristics on entrepreneurial intentions decreased but remained significant. There are no studies in the literature to support this finding.

This study presents some important results for individuals, businesses and countries, and brings some suggestions together. It is important to identify potential entrepreneurial candidates who can create economic value for their countries and businesses and those who intend to become entrepreneurs in the future. Identifying, supporting, and encouraging students and employees who intend to become entrepreneurs is necessary in order to provide a sustainable competitive advantage. On the other hand, it is important to determine the personality characteristics of those candidates who have entrepreneurship intentions and to investigate which individuals with these characteristics could be more entrepreneurial. Through determining the personality characteristics of entrepreneurial individuals, education should be provided to improve these characteristics in children, starting from primary school. It is also important to educate individuals who can access information and use it to create new knowledge in the philosophy of lifelong learning, and to play an important role in educational institutions.

Similarly, countries' increasing needs for individuals who are innovative, creative, and able to think differently, as a result of which they can turn their thoughts into activities, contributing both to themselves and to their countries, have highlighted the concept of entrepreneurship. In this context, in order to develop entrepreneurship, priority has been given to practices that support entrepreneurship in every institution, and especially the state. As mentioned above, the state plays a major role in identifying and training potential entrepreneurial candidates in universities. According to the findings of this research, a positive relationship was found between personality characteristics, lifelong learning, and the entrepreneurship tendencies of university students. Although there is a positive relationship between the five dimensions of personality and entrepreneurship tendencies, a stronger relationship was found in particular with the dimensions of emotional balance and openness to experience. In other words, it can be seen that individuals who are highly consistent in their relationships and emotions, and who are open to learning and trying new things, have a greater tendency to start jobs and complete the work they started. The weakest relationship was found to be extraversion; it was determined that students with active and communicative characteristics had a relatively fewer entrepreneurial tendencies than the other dimensions. In addition, the lifelong learning tendencies of students who have the personality characteristics of responsibility, consistency, and extroversion were found to be strong.

When these findings are examined, it is important that universities provide entrepreneurial training and encourage entrepreneurship by identifying personality characteristics, lifelong learning tendencies, and entrepreneurial intentions in students with the following personality characteristics:

consistency, openness to experience, innovation, and learning, and who have great potential to become entrepreneurs. At this point, it is recommended that students who avoid taking responsibility, who are inconsistent and closed to development and learning, and who do not have the potential to become entrepreneurs should not benefit from this training. In this way, teachers will be able to work with more competent and interested students and make concrete improvements. This will create benefits for the government, universities, and individuals in terms of labor, time, and cost.

The findings of this research reveal that students' characteristics, abilities, competencies, and tendencies need to formulate a desire to start jobs, to fulfill their responsibilities, and to achieve and continue their results. At this point, it is important to raise awareness about which individual aspects are superior and which are inferior. This is because competitiveness in today's business world can only be realized by individuals who know themselves, who are aware of their characteristics, and who develop those characteristics with the necessary knowledge, skills, and abilities. These findings have significant effects. It is recommended that training programs in higher education institutions are designed, especially in order to identify and develop the personality characteristics and values of business leaders who produce the social, environmental, and economic values of the future and transform them into sustainable entrepreneurs.

There were three basic limitations in this research. The first concerned the data collection method. This research was conducted using the survey method, and collecting data from a single source (the individual himself) may cause bias. This could increase the magnitude of the linear relationships resulting from the analysis. Collecting data from different sources in future studies will eliminate this limitation. Another limitation of this study came from the selected sample, which consisted of students from certain programs at Erciyes University. This made it difficult to generalize the research results. Students and entrepreneurial candidates from different programs and universities could be included in similar studies in the future. The third limitation of this study was the variables that were included. In this study, variables were collected and analyzed for the specified purpose. However, there are many variables other than personality and lifelong learning that affect entrepreneurship intentions. Other variables that were ignored in this study could be considered in future studies.

**Author Contributions:** E.A. contributed Conceptualization, Formal Analysis, Methodology and Writing Original Draft. G.K. contributed Investigation, Survey Preparation, Data Collection and Writing Review & Editing. H.K. contributed Investigation, Resource and Data Collection.

**Funding:** This research received no external funding.

**Conflicts of Interest:** The authors declare no conflict of interest.

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
