# Peer review of "The Effect of University Students’ Individual Innovation and Lifelong Learning Trends on Entrepreneurship Orientation"

_sustainability, doi:10.3390/su11226201_

Round 1

Reviewer 1 Report

The paper addresses an interesting research area, investigating education entrepreneurship in relation with university students. In my opinion the paper is very well structured and it presents a good discussion but the introductory section should contain a paragraph discussing the contributions of the study.

The conclusion section not offer too value to the reader. The authors should develop more the theoretical and practical implications of the study for the universities, teachers and students/society and and clarify better what aspects of the student's personality are the ones that have the most weight when it comes to influencing their entrepreneurial capacity.

Reviewer 2 Report

In general, the research is interesting, the hypotheses have been well established, the methods well chosen. However, the analysis of the results is very poor. It is basically based only on the analysis of the statistical results. The explanation of the statistically examined processes is too vague and therefore weak. The authors should develop the results analysis and discussion with the results of other authors' research, because it is limited only to some sentences in the conclusions.

Language errors and stylistic awkwardness should be corrected. For example, in this sentence the use of the word "but" is incomprehensible: "In the research, it was found that the participants' personality traits and sub-dimensions had a significant but positive relationship with their entrepreneurial intentions."

Since the research sample is limited to one specific university, I see little possibility of interest in the article by readers from abroad. However, the proposed research procedure can be applied to other groups and thus wider research of students from many universities.

Round 2

Reviewer 2 Report

The changes after the first review are satisfying in my opinion.

Author Response

Mr. Referee
We have completed the changes you want us to make. However, we would like to inform you that we have the help of a professional company and we have edited the article. Thank you for your contribution.
